# History-dependent domain and skyrmion formation in 2D van der Waals magnet Fe₃GeTe₂

M. T. Birch [1,8 ✉], L. Powalla [2,8 ✉], S. Wintz[1], O. Hovorka[3], K. Litzius [1], J. C. Loudon [4], L. A. Turnbull [5], V. Nehruji [3], K. Son[1,6], C. Bubeck[1], T. G. Rauch [7], M. Weigand [7], E. Goering[1], M. Burghard [2] & G. Schütz[1]

The discovery of two-dimensional magnets has initiated a new field of research, exploring both fundamental low-dimensional magnetism, and prospective spintronic applications. Recently, observations of magnetic skyrmions in the 2D ferromagnet Fe₃GeTe₂ (FGT) have been reported, introducing further application possibilities. However, controlling the exhibited magnetic state requires systematic knowledge of the history-dependence of the spin textures, which remains largely unexplored in 2D magnets. In this work, we utilise real-space imaging, and complementary simulations, to determine and explain the thickness-dependent magnetic phase diagrams of an exfoliated FGT flake, revealing a complex, history-dependent emergence of the uniformly magnetised, stripe domain and skyrmion states. The results show that the interplay of the dominant dipolar interaction and strongly temperature dependent out-of-plane anisotropy energy terms enables the selective stabilisation of all three states at zero field, and at a single temperature, while the Dzyaloshinksii-Moriya interaction must be present to realise the observed Néel-type domain walls. The findings open perspectives for 2D devices incorporating topological spin textures.

¹ Max Planck Institute for Intelligent Systems, 70569 Stuttgart, Germany. ² Max Planck Institute for Solid State Research, 70569 Stuttgart, Germany. ³ Faculty of Engineering and Physical Sciences, University of Southampton, Southampton SO17 1BJ, UK. ⁴ Department of Materials Science and Metallurgy, University of Cambridge, Cambridge CB3 0FS, UK. ⁵ Department of Physics, Durham University, Durham DH1 3LE, UK. ⁶ Department of Physics Education, Kongju National University, Gongju 32588, South Korea. ⁷ Helmholtz-Zentrum Berlin für Materialien und Energie GmbH, Institut Nanospektroskopie, 12489 Berlin, Germany. ⁸ These authors contributed equally: M. T. Birch, L. Powalla. ✉email: birch@is.mpg.de; l.powalla@fkf.mpg.de

The field of spintronics, which aims to harness the spin degree of freedom of electrons for efficient information storage and logic devices, has received a major impetus through the recent advent of two-dimensional (2D) magnets[1–5]. 2D materials offer unique spin-related properties, such as long spin relaxation time and spin diffusion length[6], or spin-valley locking[7]. Moreover, stacking individual 2D materials into van der Waals (vdW) heterostructures allows exploitation of interlayer spin-orbit or magnetic exchange proximity effects[8,9], and therefore direct design of the magnetic performance. Along these lines, numerous initial prototype devices have already been demonstrated[10–14]. The first experimentally observed 2D magnets were few-layered single crystals of the insulators $Cr_2Ge_2Te_6$[15] and $CrI_3$[16] in 2017. A range of further 2D magnets have since been discovered[17–21], including the target of the present study: metallic ferromagnet $Fe_3GeTe_2$ (FGT)[22–25]. This 2D material has attracted particular attention due to its interesting charge transport properties[26], heavy fermion behaviour[27], and high Curie temperature, $T_C$, which is increased for higher Fe content[28], and can approach room temperature through a gate voltage-controlled mechanism[29].

Recently, the formation of magnetic skyrmions—topological nanoscale whirls in the spin structure of magnetic materials[30,31]—has been reported in FGT flakes[32–38]. This discovery opens up further prospective applications such as 2D skyrmion racetrack or neuromorphic computing devices[40,41]. To this end, the possibility of exploiting the atomically flat and well-defined interfaces offered by vdW materials presents significant advantages over typical sputtered multilayer thin-film skyrmion systems[42,43]. However, there have been a variety of conflicting mechanisms proposed for skyrmion formation in FGT flakes, including the dipolar interaction[32], an intrinsic interfacial Dzyaloshinskii-Moriya interaction (DMI) between the Te and Fe layer interface[37], an interfacial DMI between pristine FGT and its naturally oxidised top and bottom layers[33,35], DMI induced by intrinsic defects within the bulk of the material[38], or more complex higher-order contributions[39]. The matter has been further complicated by the observation of both Bloch[32,35] and Néel-type[33–37] skyrmions.

Moreover, the stability and formation of skyrmions, and related spin textures, in FGT flakes has yet to be fully explored. A well-established method to investigate such stability is to explore the magnetic phase diagram for different field and temperature histories. In particular, history-dependent control of charge and spin degrees of freedom can result in the formation of quenched states, such as charge or spin glasses, which may be otherwise hidden in thermoequilibrium[44]. For the bulk skyrmion materials, such explorations led to fascinating results: while typically skyrmions only formed in a limited range of applied field and temperature close to $T_C$[30], they were discovered to exist outside this region both in a metastable state[45], and stabilised by alternative mechanisms such as magnetic frustration[46], or an anisotropic exchange interaction[47]. Therefore, we identified a compelling need for a comprehensive study to investigate the stability and history-dependent formation of magnetic skyrmions in FGT flakes, and to clarify the balance of interactions responsible for their formation.

In this work, we utilise a combination of real-space magnetic imaging methods to investigate the formation of uniform, stripe, and skyrmion states in an exfoliated flake of FGT with a range of thickness regions. Notably, we discovered that the history-dependence of the flake is such that all observed magnetic structures can be realised at a single temperature and at zero field by following specific temperature and field paths. We elucidate this behaviour by determining extensive magnetic phase diagrams following three distinct measurement protocols, revealing the temperature dependence of the skyrmion and stripe domain formation. The extent of the history-dependence is distinctly different from other known skyrmion systems, and appears to originate from the strong, temperature-dependent anisotropy present in FGT. This renders the material particularly advantageous for studying the fundamental domain and skyrmion evolution in 2D magnets, as well as for exploring the potential technological impact. Finally, with comparison to micromagnetic and mean-field simulations, we demonstrate that the combination of the dipolar interaction and out-of-plane anisotropy is likely the dominant stabilisation mechanism of the skyrmions, albeit some interfacial DMI contribution must be present to realise the observed Néel-type domain walls.

## Results

**FGT flake characterisation.** Energy dispersive x-ray spectroscopy established the initial bulk FGT crystal to be slightly Fe deficient, with an elemental composition of $Fe_{2.95}GeTe_{1.70}$ (see methods, Supplementary Note 1, Supplementary Fig. S1). SQUID magnetometry measurements (see methods) determined $T_C$ of the crystal to be 195 K, and confirmed the strong, out-of-plane, uniaxial magnetocrystalline anisotropy[22–25], which was found to roughly linearly increase with decreasing temperature, from a value of 170 kJ/m$^3$ at 200 K to 450 kJ/m$^3$ at 50 K (see Supplementary Fig. S2). The sample construction, depicted in Fig. 1a, consists of an exfoliated FGT flake stamped onto a $Si_3N_4$ membrane, and capped with a flake of hexagonal boron nitride (hBN) (see methods, Supplementary Fig. S3).

The principle of scanning transmission x-ray microscopy (STXM), which exploits x-ray magnetic circular dichroism (XMCD) to obtain the local electronic state and out-of-plane component of the magnetisation, $m_z$, is also illustrated in Fig. 1a[48]. A STXM image of the investigated FGT flake is shown in Fig. 1b, revealing a range of thickness steps between 10 and 60 nm, as determined by atomic force microscopy measurements (see Supplementary Note 2, Supplementary Fig. S4). By measuring x-ray absorption spectra (see methods) for both bulk and flake samples, and performing XMCD sum rules analysis we determined the magnetic moment to be 1.52 $\mu_B$/Fe, and estimated that ~10 nm of both surfaces of the flake sample were oxidised before being capped by the hBN layer, resulting in the regions thinner than ~20 nm exhibiting no resolvable ferromagnetic domain ordering (see Supplementary Note 3 and Supplementary Figs. S5, S6). This estimation was supported by real-space transmission electron microscopy measurements of a similarly prepared sample stack, which indicated an oxide layer thickness of ~ 7 nm at both surfaces of the FGT flake (see Supplementary Fig. S7). Therefore, we focused our STXM measurements on the region of interest (ROI) indicated, which exhibits three thickness regions: 60, 50 and 35 nm.

The first key result of this work—the observation of the substantial history-dependence of the magnetic state of the FGT flake—is apparent when considering Fig. 1c–f. By following different field and temperature protocols, illustrated in Fig. 1c, five distinct magnetic states could be selected at 150 K and 0 mT. By zero-field cooling (ZFC) from above $T_C$, 186 K in the flake sample, a labyrinth-like stripe domain (SD) state was achieved, as shown in Fig. 1d. Following the field sweep (FS) protocol, by initialising the sample at ±250 mT and then setting the desired magnetic field, a uniformly magnetised (UM) state was realised, aligned in either the positive or negative out-of-plane direction (UM+ and UM−), respectively. An example of the UM− state is shown in Fig. 1e. Finally, field-cooling (FC) the sample from above $T_C$ under an applied field between ±10–25 mT, resulted in the formation of a disordered array of skyrmions, with their cores

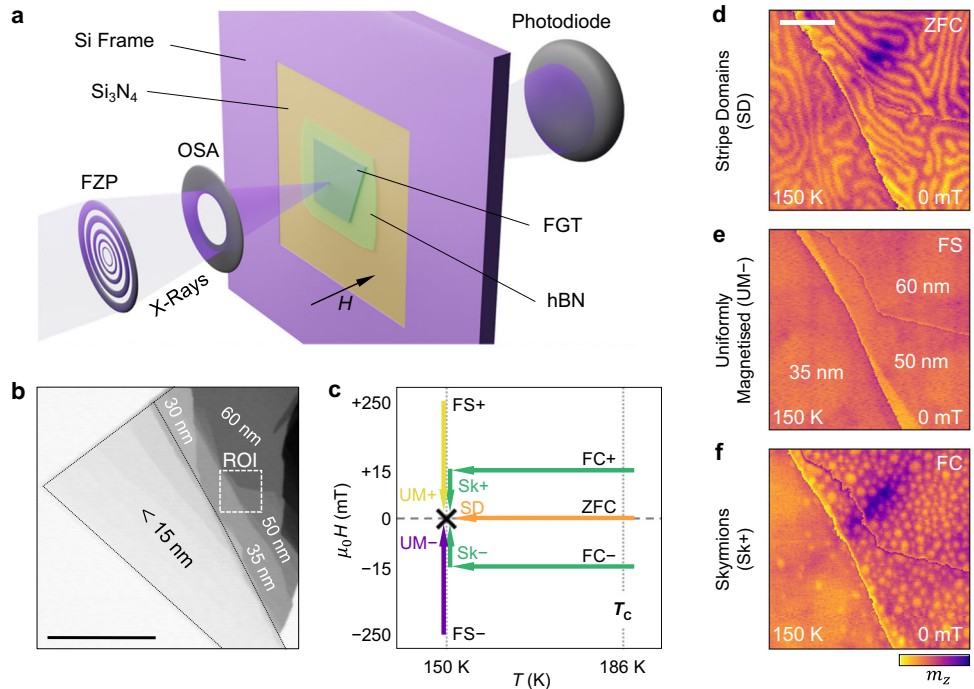

**Fig. 1 Fe₃GeTe₂ exfoliated flake sample and observed spin textures. a** Visualisation of the Fe₃GeTe₂ (FGT) and hexaboron nitride (hBN) sample arrangement and the scanning transmission x-ray microscopy (STXM) technique. The x-ray beam is focused to a ~20 nm spot size by a Fresnel zone plate (FZP) and an order selection aperture (OSA). Transmission through the sample is measured pixel by pixel by the photodiode detector. The orientation of the applied magnetic field $H$ is indicated. **b** STXM image of the FGT flake sample. Thicknesses of the different regions are indicated. The region of interest (ROI) featured in the STXM images within this work is indicated. The scale bar is 10 μm. **c** Schematic illustration of the zero field cooling (ZFC), field sweep (FS) and field cooling (FC) measurement protocols, which respectively can realise the stripe domain (SD), uniformly magnetised (UM±) and skyrmion (Sk ±) states at the same temperature and zero field (here: 150 K and 0 mT). **d–f** Magnetic contrast STXM images of the SD, UM and Sk states in the ROI at 150 K and 0 mT, realised by following the ZFC, FS and FC protocols shown in **c**, respectively. The scale bar is 1 μm.

aligned in either the negative (Sk+) or positive (Sk−) out-of-plane direction. Once formed, these skyrmions persisted after reducing the field to 0 mT, as featured in Fig. 1f.

**Magnetic phase diagrams**. To explore the history dependence of the magnetic states further, we performed detailed STXM imaging of the ROI following the three measurement protocols, resulting in the magnetic phase diagrams displayed in Fig. 2. The measurement paths are illustrated by the respective arrows in Fig. 2a–c. Points where the SD and skyrmion states coexisted were included in the skyrmion region of the phase diagram for clarity. Looking first at the results of ZFC measurements in the 60 nm region, shown in Fig. 2a, the magnetic phase diagram is symmetric about 0 mT, and the SD state existed from 186 K down to the instrument's base temperature of 30 K. Interestingly, the skyrmion state was only observed in a small range of temperatures and applied fields close to $T_C$. This behaviour differs from typical materials hosting dipolar-stabilised skyrmions, such as perovskites or garnets, where they typically form over a much larger temperature range[49].

The magnetic phase diagram was found to be significantly different when following the FS protocol, as shown in Fig. 2b. In particular, below 150 K formation of the SD state was not observed—the entire flake sample uniformly switched between the UM+ and UM− states. Such a crossover between multi-domain and monodomain behaviour has been argued based on magnetoresistance measurements in previous studies of FGT[22], and here we provide direct proof of its existence. Furthermore, we observed an asymmetry in the formation of the skyrmion state: at 180 K, skyrmions only emerged from the SD state at positive

fields, while at 185 K, they also nucleated directly from the UM− state at negative fields.

Finally, Fig. 2c shows the phase diagram acquired via the FC procedure with a cooling field of 15 mT. This demonstrates the possibility to realise the skyrmion state over a large portion of the magnetic phase diagram by quenching the skyrmions formed close to $T_C$ down to lower temperatures. It is interesting to note that the skyrmions are stable at 0 mT, and that skyrmions formed with a positive applied field appear to survive to larger negative fields, and vice versa. The field asymmetry of the presented FS and FC phase diagrams is only a feature of the specific path followed during the measurements: if the FS phase diagram were taken with increasingly negative field, or if the FC phase diagram were acquired by field cooling at a negative field, the field asymmetry would be reversed.

For the 50 nm and 35 nm thicknesses, plotted in Fig. 2d–i, the phase diagrams are qualitatively similar to those of the 60 nm region, although there are two major differences. Firstly, the value of $T_C$ appeared to be reduced to 185 K and 180 K in the 50 and 35 nm regions, respectively. This is consistent with other reports of decreased $T_C$ in thin flake samples of FGT[22], and would place the thicknesses of the magnetic regions of our sample around 10–40 nm, as expected from the oxide layer thickness estimation. Secondly, the saturation field was reduced with decreasing thickness, resulting in the smaller field range of the SD and skyrmion states in thinner regions. The phase diagrams reveal the high tunability of the magnetic state at low temperatures, and the significant impact of the thermal and field history of the FGT flake. Both the SD and skyrmion states can be realised over a large range of temperatures and fields via a heating and cooling process. However, once the spin textures are annihilated by an applied field below 150 K, only the

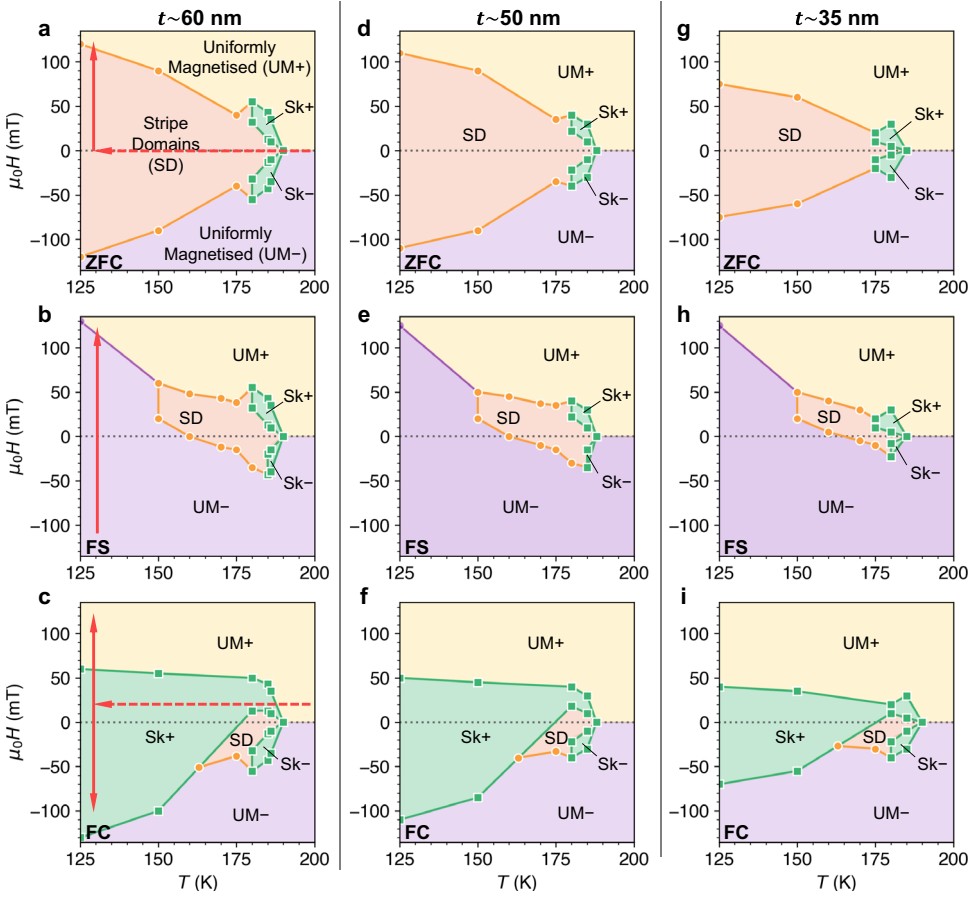

**Fig. 2 History and thickness dependent magnetic phase diagrams.** Magnetic phase diagrams determined by imaging the $Fe_3GeTe_2$ (FGT) flake as a function of temperature and applied field for three different thickness regions: 60 nm (**a–c**), 50 nm (**d–f**) and 35 nm (**g–i**). Phase diagrams were determined by following the zero field cooling (ZFC) (**a**, **d**, **g**), field sweep (FS) (**b**, **e**, **h**), or field cooling (FC) at 15 mT (**c**, **f**, **i**) measurement procedures. The measurement paths are indicated by the red arrows in **a–c**. The existence of the stripe domain (SD, orange), skyrmion (Sk±, green) and uniformly magnetised (UM+, yellow, UM−, purple) states are indicated by the coloured regions. Measured boundary points are indicated by the markers (orange circles for SD, green squares for Sk states).

uniformly magnetised states can be selected without subsequent temperature cycling. Such state switching could offer opportunities for non-volatile phase-change memory functions[44].

The observed magnetic phase diagrams are quite different from those found in other skyrmion systems, such as bulk chiral magnets and chiral multilayer thin films. Specifically, the limited temperature range of skyrmion formation observed in our ZFC and FS measurements is reminiscent of the small skyrmion pocket seen in bulk chiral magnets[30,31]. However, it is interesting to note that in thin lamellae of these B20 systems, the skyrmion pocket typically expands to lower temperatures with decreasing thickness[50] – quite different to the behaviour observed here in FGT flakes. Due to the focus on room temperature applications, there have been few investigations into the temperature-field magnetic phase diagrams of chiral multilayer thin film systems. However, here the main difference is that, similarly to the centrosymmetric non-chiral skyrmion materials[49], skyrmions in multilayers are typically able to form at a large range of temperatures[51]. Most notably, while it is common to be able to stabilise both stripe and skyrmion states at 0 mT in all these systems, the stabilisation of a uniformly magnetised state at zero field has not been observed. As shall be explored in our simulations, it is likely that the unique, history-dependent magnetic phase diagrams exhibited by FGT flakes are due to the increasingly hard magnetic properties of FGT flakes with decreasing temperature.

**Real-space imaging of magnetic textures**. Closer inspection of the real-space data reveals clues about the stabilisation mechanism of the observed spin textures. Selected STXM images acquired at different temperatures following the FS protocol are shown in Fig. 3a–t (details of contrast normalisation and further STXM data are shown in Supplementary Note 4, Supplementary Figs. S8–10). During each field sweep, the formation of the stripe domain state appears to be precipitated by an initial local nucleation of the oppositely magnetised stripe domain, which proliferates through the rest of the flake to form a labyrinth-like domain state. At higher temperatures and with increasing field, these stripe domains are broken up into individual skyrmions. The density of skyrmion formation is strongly temperature dependent in the vicinity of $T_C$: At 185 K, in Fig. 3p, q, s, t, skyrmions fill the 60 nm and 50 nm thickness regions of the sample. In comparison, at 180 K (Fig. 3k, l), only a few isolated skyrmions are found in the same thickness regions. Due to the reduced $T_C$ of the 35 nm region, similar behaviour was observed in the 175 K and 180 K datasets, in Fig. 3h, n.

Lorentz transmission electron microscopy (LTEM) of a comparable FGT flake was performed to investigate the character of the domain walls (see Supplementary Note 5). Figure 3u–x shows the walls becoming more widely spaced and then disappearing as the applied magnetic field is increased following the ZFC procedure at 92 K. The change in image contrast by tilting by an angle $\alpha$, shown in Fig. 3x, y indicates that our FGT

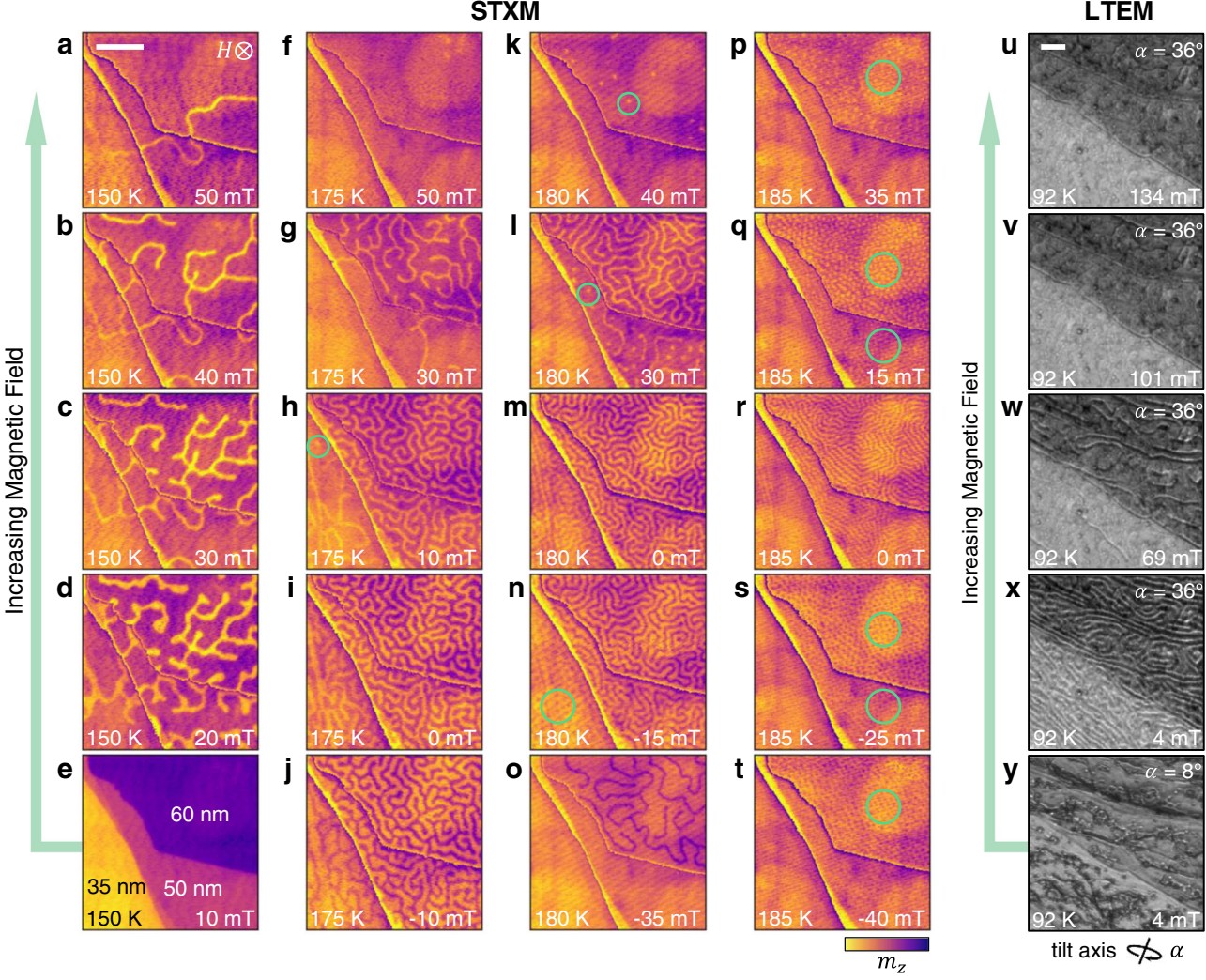

**Fig. 3 Real-space imaging of spin textures. a–t** Magnetic contrast scanning transmission x-ray microscopy (STXM) images of the FGT flake region of interest measured following the field sweep protocol from −250 to 250 mT at a range of temperatures. In each image, the contrast of each thickness region has been normalised to highlight the magnetic structures (original thickness contrast without domains can be seen in panel (**e**)). The colourmap indicates the signal proportional to the out-of-plane magnetisation $m_z$. Regions containing skyrmions are highlighted with a green circle. **u–y**, Lorentz transmission electron microscopy (LTEM) images acquired with increasing field after zero-field cooling. The sample tilt angle $\alpha$ is labelled. The scale bars are 1 μm.

samples exhibit Néel-type domain walls rather than Bloch-type (see Supplementary Figs. S11, S12), as observed in previous studies of thin flake samples[33,35].

The real-space measurements also reveal that the average size of the stripe domains and skyrmions is increased at lower temperatures[52]. This result is quantified in Fig. 4a, where the average stripe domain size at 0 mT is plotted as a function of temperature (see Supplementary Note 6, Supplementary Figs. S13, S14). In addition, there is an indication that, for a fixed temperature, the thinner regions exhibit slightly larger domain sizes. The inset shows that a similar trend is seen for the average skyrmion diameter. The average domain wall width at 0 mT was also extracted, and is plotted as a function of temperature in Fig. 4b, showing a clear increase with decreasing temperature.

These results have important implications concerning the origin of the SD and skyrmion states. The observed temperature and thickness dependence of the domain size could be explained by the interplay of the dipolar interaction with the temperature-dependent uniaxial anisotropy observed in the magnetometry measurements. On the other hand, since Bloch-type domain walls are typically the lower energy configuration in out-of-plane

magnetised films[53], the LTEM observation of Néel-type domain walls is a strong indication that there must be some DMI contribution present to twist the SD and skyrmion helicity.

**Skyrmion field stability.** To investigate the stability of the skyrmions, we studied their field evolution in detail, with the results shown in Fig. 5a–i. First, the FGT flake was initialised following the FC protocol. As the sample passes below $T_C$ under an applied field, the interplay of the Zeeman and dipolar interactions results in the formation of a dense disordered array of skyrmions, which survives down to low temperatures beyond their formation region, as shown for a FC field of 15 mT in Fig. 5b. We utilised image recognition software to extract the number and diameter $d$ of the skyrmions in each image (see Supplementary Note 7, Supplementary Figs. S15, S16). These quantities have a strong dependence on the cooling field, as shown by the plots in Fig. 5f and g. For the 35 nm region, skyrmions were observed only when applying cooling fields of 15 mT and below.

After the 15 mT FC initialisation, images were acquired for both increasing and decreasing applied field, with the results shown in

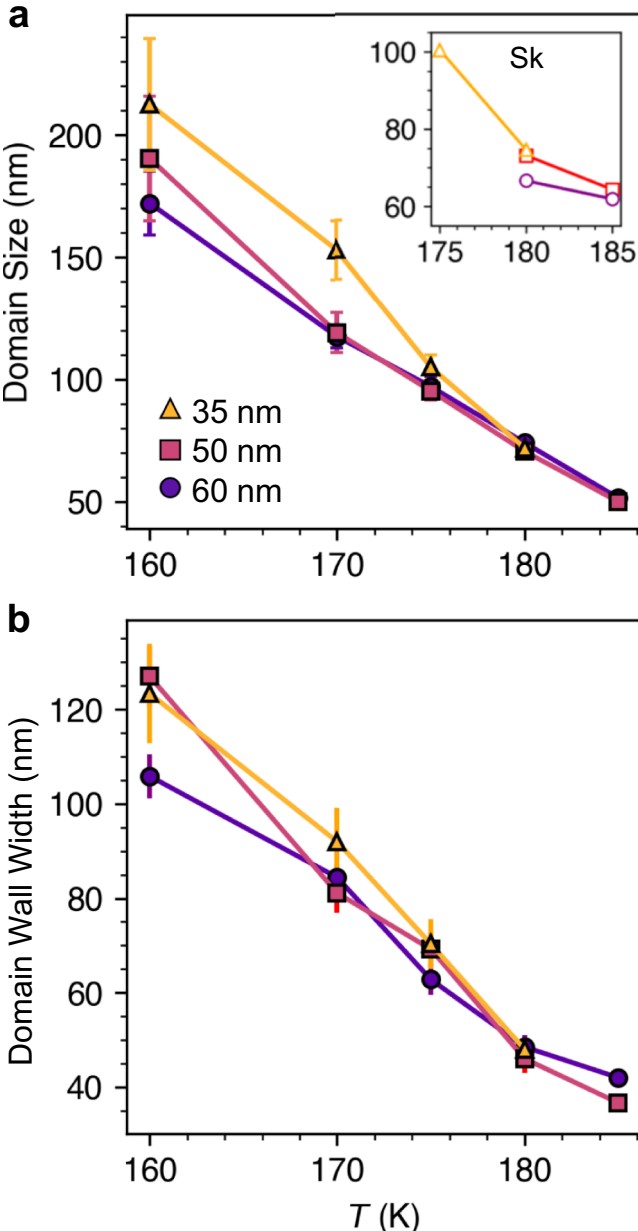

**Fig. 4 Domain size and domain wall width as a function of temperature and sample thickness. a** The stripe domain size, determined from the x-ray microscopy images at 0 mT following the field sweep protocol, as a function of temperature for each sample thickness. The inset plots the average diameters of the skyrmions (Sk) over all fields as a function of temperature for each sample thickness. **b** The domain wall width of stripe domains determined at 0 mT as a function of temperature for each sample thickness. Error bars indicate the standard error of all domains sizes/widths averaged at each field and temperature point.

Fig. 5a–e. The number and $d$ of the skyrmions in each thickness region during the field sweep are plotted in 5h and i. For increasing fields, the number of skyrmions quickly decreases, while the average size of the skyrmions slightly decreases. For decreasing field, the number of skyrmions also decreases, while $d$ strongly increases. The filled coloured regions indicate the minimum and maximum diameter of the observed skyrmions at each field, showing that the increasing average size is largely driven by an expansion of a subset of the skyrmions, while the smallest skyrmions remain at a similar $d$, perhaps due to pinning effects.

In their theoretical treatment of skyrmion formation, Büttner et al. laid out useful indicators to distinguish between DMI and dipolar stabilised skyrmions[54]. The model predicts that DMI-stabilised skyrmions should not significantly alter their size as a function of the applied magnetic field, whereas skyrmions predominantly stabilised by the dipolar interaction can be expected to be sensitive to such field changes. Therefore, the present observations provide a strong indication that the dipolar interaction plays a dominant role in the stabilisation of the observed skyrmion states in FGT.

**Comparison to magnetic simulations**. To validate this conclusion, we attempted to compare the experimental data with the results of micromagnetic simulations, utilising realistic values for the micromagnetic parameters (see methods). To determine the approximate value of interfacial DMI, $D$, required to form Néel-type domain walls in the system, we simulated skyrmion states with a range of $D$ values and observed the helicity of the resulting magnetic textures. As shown in Fig. 6a–d, we found that, for our chosen micromagnetic parameters approximating the FGT flake, a value of $D = 0.12$ mJ/m$^2$ was sufficient to alter the domain wall helicity from the expected Bloch-type to the experimentally observed Néel-type. The particular value will depend on factors such as the values of the other micromagnetic parameters, and the sample thickness. Selecting this parameter, we modelled the experimental FC results observed in Fig. 5a–e by nucleating a disordered array of skyrmions at 20 mT, and simulated field sweeps with both increasing and decreasing field, as shown in Fig. 6e. Similarly to the experiment, the skyrmions shrink for increasing field, and grow and form patch-like structures for negative magnetic fields—typical signs of a predominantly dipolar-stabilised spin texture[54].

Since it is challenging to model temperature with micromagnetic simulations, we performed complementary mean-field calculations to study the temperature-dependent domain evolution observed in the field-sweep measurements in Fig. 3 (see methods). Optimal mean-field model parameters were tuned such that the field sweeps produced domain structures comparable to the experiment, and were estimated to be consistent with micromagnetic values (see "Methods", Supplementary Note 8). In order to recreate the experimentally observed temperature dependence of the SD and skyrmion formation, it was necessary to scale the uniaxial anisotropy, $K$, with respect to the saturation magnetisation. Utilising a classical Callen-Callen power law scaling of $K$, the model captured the high-temperature behaviour well, but did not show the transition to monodomain switching at low temperature (see Supplementary Fig. S17). However, inspired by the large increase in uniaxial anisotropy at low temperatures observed in the magnetometry measurements, we included an additional temperature-dependent prefactor to the scaling law to further increase the anisotropy at lower temperatures (See Supplementary Note 9 and Supplementary Fig. S18). Together with an increase in the exchange interaction at lower temperatures, the model qualitatively reproduces the experimental FS phase diagram, including additional details such as the observed increase in domain size and crossover to monodomain behaviour at lower temperatures, as shown in Fig. 6g–i.

## Discussion
The combination of real-space imaging and simulation methods allows us to clarify the origin of skyrmion formation in FGT flakes. Specifically, the observed temperature dependence of the typical stripe domain size, and the field dependence of the skyrmion diameters, indicates that the spin textures are primarily stabilised by the dipolar interaction. The mean-field simulations

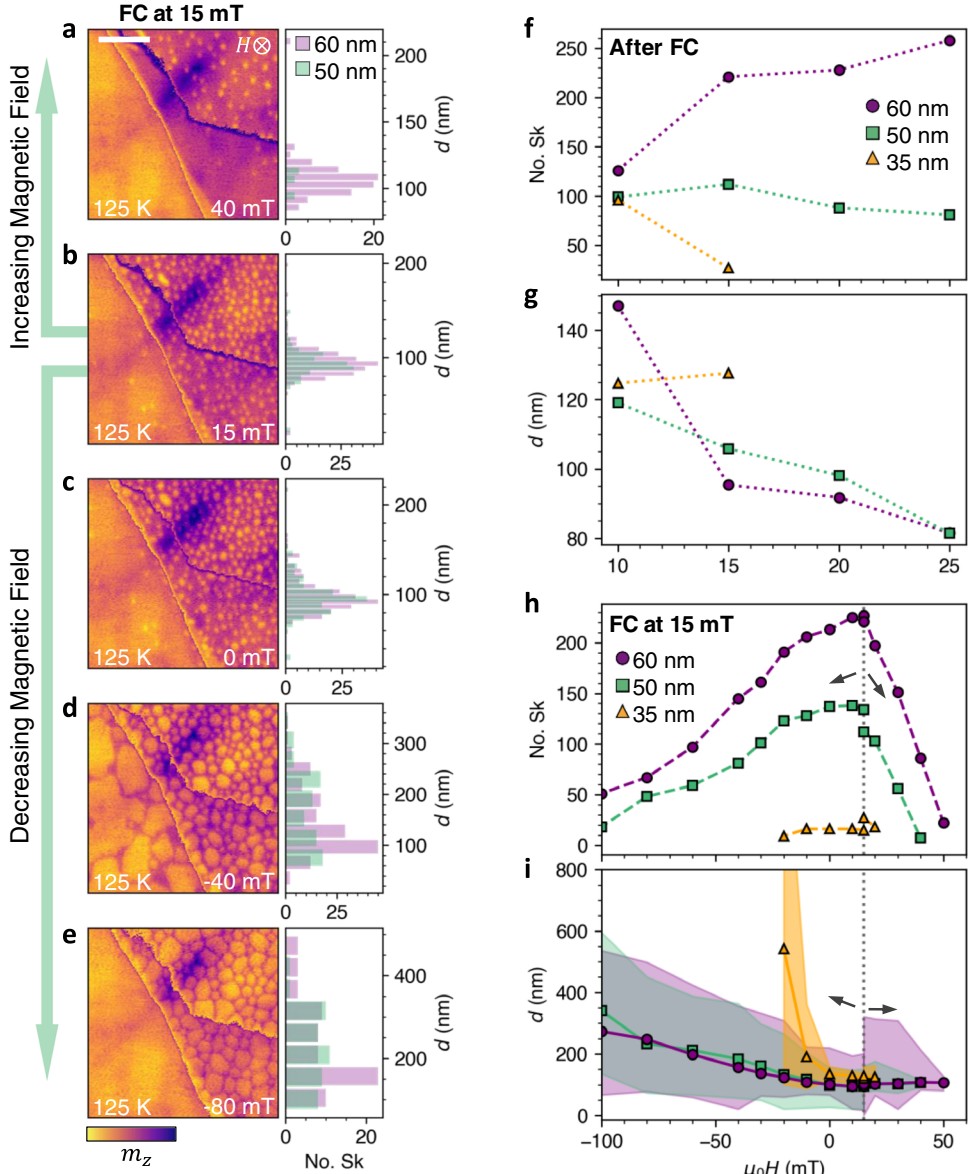

**Fig. 5 Formation of skyrmions by field-cooling. a–e** Scanning transmission x-ray microscopy (STXM) images of the Fe$_3$GeTe$_2$ (FGT) flake acquired as a function of applied magnetic field after field-cooling (FC) the sample from 200 to 125 K under an applied field of 15 mT. The colourmap is proportional the normalised out-of-plane magnetisation of each thickness region. The scale bars are 1 μm. The histograms plot the distribution of the skyrmion size in the 60 nm (purple) and 50 nm (green) thickness regions. **f**, **g** The number and diameter, $d$, of skyrmions determined after FC at a range of applied magnetic fields in the 60 nm (purple), 50 nm (green) and 35 nm (orange) thickness regions. **h**, **i** The number and diameter of the skyrmions determined for increasing and decreasing magnetic field at 125 K, after FC the sample at 15 mT, for each thickness region. Dashed vertical lines and arrows indicate the initial point, and the direction, of the field sweeps. The shaded regions indicate the maximum and minimum $d$ observed at each field point.

support this conclusion, demonstrating that the temperature-dependent formation of skyrmions and SDs observed in the FGT flakes can be well-described by the large increase in the out-of-plane anisotropy with decreasing temperature. However, the LTEM observation of Néel-type domain walls necessitates the presence of some form of DMI contribution, although this is not required to be large. The micromagnetic simulations estimate a lower bound for the strength of the DMI in our sample, with the value of 0.12 mJ/m$^2$ being much lower than the theoretical maximum value from previous DFT calculations of oxidised FGT, which suggested values could be as high as 2 mJ/m$^2$ or more[33]. While the value we present is only a lower bound estimation, our results show that the history-dependent behaviour of FGT can be largely explained by considering the dominant dipolar and anisotropy effects. In the future, investigating a wider range of

thicknesses, including down to the monolayer, may yield deeper insights into the spin texture formation, and reveal whether skyrmions can be stabilised at the single-layer limit.

We note that observations of Bloch-type skyrmions in FGT have exclusively been in thicker (>150 nm) samples[32,35]. With this in mind, we tentatively suggest that in such thick samples, any present interfacial DMI from the oxide layer may no longer be sufficient to stabilise Néel-type domain walls, and this would provide strong indication that the DMI observed in FGT flakes must be due to some interfacial effect. However, recent studies have suggested that the DMI could instead be due to some intrinsic property of the bulk material, either due to vacancy driven DMI[38] or additional higher order effects[39]. Further work may be required to elucidate the true origin of the monochiral Néel-type domain walls. The issue of distinguishing between

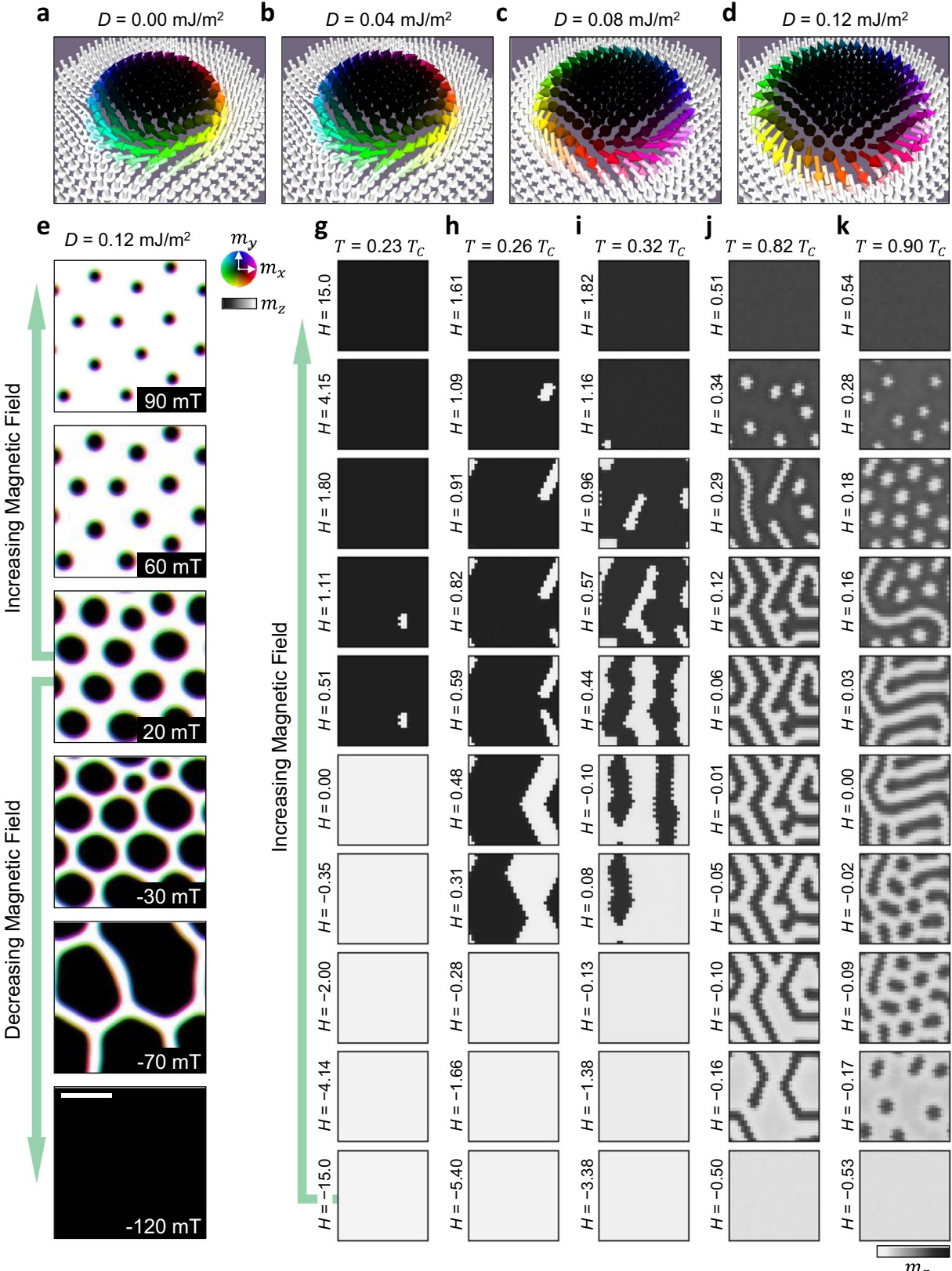

**Fig. 6 Micromagnetic and mean-field simulations. a–d** Visualisations of micromagnetic simulations showing the change in skyrmion helicity from Bloch-type to Néel-type with increasing Dzyoloshinskii-Moriya interaction strength, *D*. **e** Visualisations of micromagnetic simulations showing the evolution in size and shape of skyrmions initialised at 20 mT for both increasing and decreasing magnetic field. The scale bar is 100 nm. **g–k** Mean-field simulations of the magnetic domain structure evolution following the field sweep procedure for increasing applied field, *H*, at different temperatures, utilising the model with increased anisotropy scaling at lower temperatures. The colourwheel and colourbars indicate the in-plane and out-of-plane magnetisation, respectively.

dipolar-stabilised and DMI-stabilised skyrmions can be considered controversial because it has been known for decades that the dipolar interaction and a strong uniaxial anisotropy can stabilise magnetic bubbles[55]. Due to their topological equivalence, these objects have thus been referred to as skyrmion bubbles[56]. However, the identification of dipolar interaction-stabilised skyrmions need not imply inferiority, since their discovery in new materials, particularly in 2D magnets, offers both renewed fundamental interest and fresh possibilities for their technological implementation. Further work will uncover whether similar history-dependent behaviour may be observed in other skyrmion-hosting 2D magnets. Moreover, the future discovery and investigation of new 2D magnetic materials, such as those proposed to possess chiral crystal structures[57], or 2D multiferroics exhibiting electric field-tunable DMI[58,59], may allow for the observation of further novel topological spin textures in 2D materials.

In summary, we utilised magnetic imaging techniques to explore the formation of magnetic stripe domains and skyrmions in an exfoliated FGT flake. We determined thickness-dependent magnetic phase diagrams following three distinct measurement protocols, revealing significant differences to other skyrmion hosts, such as bulk chiral and multilayer systems, evidenced by the possibility to selectively stabilise the stripe domain, skyrmion and uniform states at the same temperature and field point. Comparison of the real-spaces images and simulations suggests that the interplay of the dipolar interaction with the temperature-dependent uniaxial anisotropy is the primary stabilisation mechanism of the observed spin textures, while a DMI contribution must be present to twist the chirality of the domain walls from Bloch-type to the observed Néel-type. These results demonstrate the possibility to alter the properties of skyrmions in FGT by exploiting both DMI and dipolar stabilisation mechanisms, which can be readily controlled in 2D magnets thanks to their advantageous stacking properties[34,36], opening the path towards advanced skyrmion-based spintronic devices composed of 2D magnet heterostructures.

## Methods

**Sample preparation and characterisation.** The $Fe_3GeTe_2$ (FGT) single crystal was sourced from a commercial company, HQ Graphene. EDX measurements were performed with a Zeiss SEM Gemini-500, equipped with a Bruker XFlash 6-60 detector. Magnetometry measurements were carried out with a Quantum Design MPMS3 vibrating sample magnetometer, where the bulk FGT crystal was aligned along the relevant crystal axis and fixed to a quartz glass rod using GE varnish. Temperature and applied magnetic field were controlled by the built-in helium cryostat. The FGT flake sample for the STXM experiments was prepared by an all-dry viscoelastic transfer method. In the first step, the FGT bulk crystal was mechanically cleaved and exfoliated onto a PDMS stamp. A stepped flake with a thickness range between 20 and 200 nm, as estimated from the optical contrast, was selected and stamped onto a 100 nm thick SiN membrane. The flake was then capped by an exfoliated hexagonal boron nitride (hBN) sheet with a thickness of ~15 nm. The whole process was performed under ambient conditions, with each side of the flake being exposed to the atmosphere for approximately 30 min. Using a Bruker Dimension Icon atomic force microscope, measurements were performed on the capped FGT flake to determine the thickness of each region. Since the hBN capping complicates the thickness evaluation of the FGT sheet, the FGT thickness was estimated by averaging over 2–5 μm long steps. The thicknesses of all flake regions were then calibrated using measurements of the x-ray transmission (see Supplementary Note 2). For the preparation of the cross-sectional transmission electron microscope (TEM) sample, a focused ion beam (FIB) instrument (FEI FIB Scios) was used, operated at 30 kV for the initial cuts and subsequently reduced to 2 kV for thinning and cleaning. The transmission electron microscopy (TEM) investigation was conducted with a JEM-ARM200F (JEOL) microscope, operated at 200 kV and equipped with a cold field emission gun and a Cs-image aberration corrector.

**Total electron yield x-ray absorption spectroscopy.** X-ray absorption spectroscopy measurements were performed on bulk FGT single crystals using the WERA beamline, at the KARA synchrotron in Karlsruhe Institute of Technology, which has an energy resolution of $\Delta E/E = 2 \times 10^{-4}$. We utilised a superconducting magnet end station providing ultra-fast field switching with field ramping rates of

1.5 T/s. The FGT samples were glued to a conductive Mo sample holder, and electrical contact was ensured by coating the edges of the sample in carbon paste. A steel rod was then glued to the top of one of the crystal samples to enable in-vacuum cleaving. The samples were mounted inside the instrument, and the steel rod was removed, cleaving the FGT crystal sample within the vacuum chamber. Cooling was achieved with a liquid nitrogen cryostat. Absorption spectra were then measured over the oxygen $K$ edge and the Fe $L_3$ and $L_2$ edges with fixed x-ray helicity. The energy was varied by controlling the monochromator at a speed of 0.2 eV/s to ensure no energy broadening. Meanwhile, the total electron yield (TEY) from the sample and incident x-ray intensity ($I_0$ value) were measured using a Keithley 6517A electrometer. No noticeable energy drifts were observed between consecutive spectra.

**Scanning transmission x-ray microscopy.** Scanning transmission x-ray microscopy (STXM) measurements were performed with the MAXYMUS instrument on the UE46 beamline at the BESSY II electron storage ring operated by the Helmholtz-Zentrum Berlin für Materialien und Energie. With the sample mounted inside the microscope, cooling was achieved by a He cryostat and the applied magnetic field was controlled by varying the arrangement of four permanent magnets. The x-ray beam was focused to a 20 nm spot size using a Fresnel zone plate and order selection aperture, setting the approximate spatial resolution. This focused beam, with a nominal x-ray energy of 707.5 eV, was then rastered across the sample pixel by pixel using a piezoelectric motor stage. By exploiting the effects of XMCD at the resonant x-ray energy at the Fe $L_3$ edge, the transmission of the sample at each point was measured to form an image of the non-magnetic and magnetic domain structure, with the magnetic signal proportional to the out-of-plane magnetisation $m_z$. The presented images of the magnetic domain structure were recorded using a single circular x-ray polarisation. Photons were counted by an avalanche photodiode.

**Lorentz electron transmission microscopy.** LTEM measurements were performed using an FEI Titan[3] transmission electron microscope equipped with a field-emission electron gun and operated at an acceleration voltage of 300 kV. In normal operation, the electromagnetic objective lens applies a 2 T field to the specimen which would force it into the saturated state. Instead, images were acquired in low-magnification mode where the image is formed using the diffraction lens and the objective lens weakly excited to apply a small magnetic field parallel to the electron beam. The applied field corresponding to a given objective lens current was calibrated to within 1 mT using a Thermo Fisher Hall probe holder. The specimen was cooled in-situ using a liquid nitrogen cooled Gatan double-tilt specimen holder which has a base temperature of 90 K. The FGT sample investigated by LTEM exhibited thickness regions between 15 and 70 nm.

Images were energy-filtered using a Gatan 865 Tridiem so that only electrons which had lost between 0 and 10 eV upon passing through the specimen contributed to the image, and an aperture subtending a half-angle of 0.14 mrad was used to ensure that only the 000 beam and the beams associated with magnetic scattering contributed to the image. Images were recorded on a 2048 by 2048 pixel charge-coupled device (CCD). The defocus and magnification were calibrated by acquiring images with the same lens settings from Agar Scientific's S106 calibration specimen which consists of lines spaced by 463 nm ruled on an amorphous film. The defocus was found by taking digital Fourier transforms of these images and measuring the radii of the dark rings that result from the contrast transfer function of the lens[60].

**Micromagnetic simulations.** Micromagnetic simulations following the Landau-Lifshitz-Gilbert equation were performed using the MicroMagnum framework with custom extensions for the DMI. The simulations are based on the experimentally measured $M_s = 250$ kA/m, and utilised the real-space images of stripe domains to estimate the uniaxial anisotropy $K = 44.2$ kJ/m$^3$ for the thin FGT flake and the exchange interaction $A = 0.7$ pJ/m. The anisotropy determined by SQUID measurements for a bulk crystal appears to significantly overestimate the value for a thin exfoliated flake, perhaps due to the enhanced shape anisotropy, which made this adjustment necessary. The system was simulated in two configurations, both with a cell size of $2 \times 2 \times 20$ nm$^3$. For Fig. 6a–d, the disk geometry was simulated by $50 \times 50 \times 1$ cells, with only cells within a radius of 25 cells active. For the DMI estimation, varying DMI values were tested in 0.01 mJ/m$^2$ steps until the Néel configuration was realised at $D = 0.12$ mJ/m$^2$. For Fig. 6e, the system, with dimensions of $150 \times 150 \times 1$ cells, was initialised with a disordered array of skyrmions that were relaxed into their equilibrium state at 20 mT. From this initial state, the field was both increased and decreased, with the system relaxed at each field point.

**Mean-field simulations.** To perform temperature-dependent simulations we developed a mean-field model based on the standard classical spin Hamiltonian[61] with spins distributed on a two-dimensional hexagonal lattice of $30 \times 30$ spins (approx. 150 nm × 150 nm) with periodic boundary conditions. The mean-field

energy reads:

$$\mathcal{H}_{\mathrm{MF}} = -\frac{1}{2}J_{\mathrm{E}}(T)\sum_{\langle ij \rangle}\mathbf{m}_i \cdot \mathbf{m}_j - \frac{1}{2}J_{\mathrm{D}}(T)\sum_{\langle ij \rangle}\mathbf{d}_{ij} \cdot (\mathbf{m}_i \times \mathbf{m}_j) - J_{\mathrm{K}}(T)\sum_i (\hat{\mathbf{n}} \cdot \hat{\mathbf{m}}_i)^2$$
$$-\sum_i \mathbf{m}_i \cdot \mathbf{B} - \frac{1}{2}J_{\mathrm{DP}}\sum_{ij}\left(-\frac{\mathbf{m}_i \cdot \mathbf{m}_j}{r_{ij}^3} + 3\frac{(\mathbf{m}_i \cdot \hat{\mathbf{r}}_{ij})(\mathbf{m}_j \cdot \hat{\mathbf{r}}_{ij})}{r_{ij}^3}\right) \qquad (1)$$

where the symbols $\mathbf{m}_i$ represent mean-field spins, and the individual energy terms starting from the left correspond to exchange, DMI, uniaxial anisotropy, Zeeman energy, and finally the dipolar energy (see Supplementary Note 8). It is worth pointing out that by taking the small angle approximation and zero temperature limit, one reduces the above model to the standard micromagnetic energy. Supplementary Note 8 identifies the relationship between the model constants $J_{\mathrm{E}}(T=0)$, $J_{\mathrm{D}}(T=0)$, $J_{\mathrm{K}}(T=0)$ (in the units of Joule J) and their micromagnetic equivalents $A$ (J/m), $D$ (J/m$^2$) and $K$ (J/m$^3$). The exchange and anisotropy energy constants $J_{\mathrm{E}}(T)$ and $J_{\mathrm{K}}(T)$ are temperature-dependent, scaled following a modified Callen-Callen power law (see Supplementary Note 9). The temperature-dependence of $J_{\mathrm{D}}(T)$ is weak and will be ignored. Thus the DMI energy term depends on temperature only through moments $\mathbf{m}_i$. The DMI vector $\mathbf{d}_{ij}$ is oriented in the lattice plane and perpendicular to the line connecting two neighbouring spins $i$ and $j$. The anisotropy unit vector $\hat{\mathbf{n}}$ is oriented along the $\hat{\mathbf{z}}$-axis perpendicular to the lattice plane. The symbols enclosed in angled brackets in the exchange and DMI energy terms imply summation over the nearest neighbour spins. The spin moment $\mathbf{m}_i$ has temperature dependent magnitude normalised to vary between $|\mathbf{m}_i| = \pm 1$ according to the expression:

$$\mathbf{m}_i = \mathcal{L}(\beta|\mathbf{B}_i^{\mathrm{e}}|)\hat{\mathbf{B}}_i^{\mathrm{e}} \qquad (2)$$

where $\mathcal{L}(x) = \coth x - x^{-1}$ is the Langevin function, $\beta = (k_{\mathrm{B}}T)^{-1}$, $k_{\mathrm{B}}$ is the Boltzmann constant, and $T$ the temperature. The vector $\mathbf{B}_i^{\mathrm{e}}$ represents the effective field acting on the spin $\mathbf{m}_i$ and $|\mathbf{B}_i^{\mathrm{e}}|$ is the magnitude of $\mathbf{B}_i^{\mathrm{e}}$. The expression for the effective field can be obtained from Eq. (1) by calculating the variational derivative $\mathbf{B}_i^{\mathrm{e}} = -\delta\mathcal{H}_{\mathrm{MF}}/\delta\mathbf{m}_i = J_{\mathrm{E}}(T)\sum_j\mathbf{m}_j - J_{\mathrm{D}}\sum_j\mathbf{d}_{ij}\times\mathbf{m}_j + 2J_{\mathrm{K}}(T)(\hat{\mathbf{n}}\cdot\hat{\mathbf{m}}_i)\hat{\mathbf{n}} + \mathbf{B} + J_{\mathrm{DP}}\sum_j(\mathbf{m}_j + 3(\mathbf{m}_j\cdot\hat{\mathbf{r}}_{ij})\hat{\mathbf{r}}_{ij})r_{ij}^{-3}$. Thus according to Eq. (2), the mean-field spin $\mathbf{m}_i$ depends on the exchange and DMI energy couplings, material anisotropy, external and dipolar field, and also on the temperature $T$. The magnetisation structures for a given set of parameters are evaluated by minimising Eq. (1) during the field or temperature evolution starting from uniquely defined initial states[61]. The parameters required to generate the results in Fig. 6 were $J_{\mathrm{E}}(0) = 0.7$, $J_{\mathrm{D}} = 0.1$, $J_{\mathrm{K}}(0) = 2.5$, $J_{\mathrm{DP}} = 0.35$. It was also necessary to identify appropriate temperature dependencies of $J_{\mathrm{E}}(T)$ and $J_{\mathrm{K}}(T)$ (see Supplementary Note 9).

## Data availability

The scanning transmission x-ray microscopy, Lorentz transmission electron microscopy, mean-field simulation and characterisation data, generated in this study have been deposited in a Zenodo online repository[62]. Any further data and materials required to reproduce the work are available from the corresponding authors upon reasonable request.

## Code availability

Code for performing the micromagnetic and mean field simulations are available from authors upon reasonable request.

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

## Acknowledgements

We thank Helmholtz-Zentrum Berlin for the allocation of synchrotron radiation beamtime at the BESSY II synchrotron. We are grateful for beamtime at KARA, at the Karlsruhe Institute of Technology. We give thanks to the technical support of T. Reindl, A. Güth, U. Waizmann, M. Hagel and J. Weis from the Nanostructuring Lab (NSL) at the Max Planck Institute for Solid State Research. We appreciate the help of J. Deuschle, T. Heil and P. van Aken for carrying out the TEM sample preparation and measurements. The authors thank S. Moody for discussions on the magnetometry data analysis. V.N. acknowledges financial support from the UK Engineering and Physical Sciences Research Council (EPSRC) Centre for Doctoral Training under grant number EP/L006766/1. O.H. and J.L. acknowledges support from EPSRC under grant number EP/N032128/1. M.B. is grateful for support from the Deutsche Forschungsgemeinschaft (DFG) via Grant BU 1125/11-1.

## Author contributions

M.T.B., L.P., M.B. and G.S. conceived the project. L.P. fabricated and characterised the flake devices. C.B. performed the EDX measurements. M.T.B. and K.S. carried out the magnetometry measurements. M.T.B., S.W., T.R. and M.W. performed the STXM measurements at the BESSY II synchrotron. M.TB., C.B. and E.G. performed the TEY x-ray spectroscopy measurements at the ANKA synchrotron. J.C.L. acquired and analysed the LTEM data. O.H. and V.N. performed the mean-field simulations. K.L. performed the micromagnetic simulations. LAT utilised image recognition software to analyse the FC STXM data. M.T.B. and L.P. wrote the manuscript, with help from the other authors. All authors discussed the results and gave feedback on the manuscript.

## Funding

## Competing interests

The authors declare no competing interests.
