## [Peer Review File · Nature Communications]

Reviewers' Comments:

Reviewer #1:

Remarks to the Author:

The manuscript by M. T. Birch et al. reports the observation of spin textures for different magnetic fields and temperature histories in the layered ferromagnet Fe₃GeTe₂. The magnetic phase diagrams are systematically studied at varied magnetic fields and temperatures using scanning transmission x-ray microscopy (STXM) and Lorentz mode transmission electron microscopy (LTEM). The authors have demonstrated that spin textures such as uniformly magnetized, stripe domain, and skyrmion states can be selected at the same temperature and field point by the various sample thicknesses on the different measurement protocols. The major concern of the work is the stabilized spin textures by the sample history prior to measurements.

The work is scientifically sound and the high-resolution magnetic imaging experiments are solid. However, the work is somewhat consistent with previous studies [Refs. 22 and 37] that has found stripe domain or skyrmion structures in layered Fe₃GeTe₂ when the film reaches certain thickness. In addition, the field cooling process (quenching process) could enhance skyrmion phase, as has already discovered in the conventional chiral-lattice magnets [Nat. Phys. 12, 62 (2015), Nat. Mater. 15, 1237 (2016), Nature 564, 95 (2018)] as well as the same layered ferromagnet (Fe₃GeTe₂) [Refs. 32-37]. So, what is the significant novelty of the history-dependent spin textures in layered Fe₃GeTe₂ compared to that of the conventional chiral-ferromagnetic multi-layers?

The authors also mentioned that the history-dependent spin textures based on the various sample thicknesses suggest that the interplay of the dipolar interaction is the primary stabilization of mechanism for the spin textures in the samples, despite the presence of the interfacial DMI. I think that the spin textures could be stabilized by the correlation among the dipolar interaction, the interfacial DMI, and the uniaxial anisotropy [Refs 49 and 50]. This correlation should be strongly affected by the sample thickness and history measurements. However, there is a weak discussion on the mechanism of spin texture formations during the different field- and temperature-histories. The authors need to discuss in detail the comparison of the spin texture formation mechanisms, which have different thicknesses and measurement-histories. Moreover, it would be useful to discuss how dipolar interaction dominantly affect the spin texture during the field cooling process. I also wonder what role the interfacial DMI has on the observed spin textures for different field and temperature histories. The thickness- and history-dependences of micromagnetic simulations can be considered as a convincing argument supporting the role of dipolar interaction in the layered Fe₃GeTe₂. Furthermore, what is the origin of the asymmetry in the formation of the skyrmion states via the field cooling process as shown in Fig. 2? I also wonder how the thicknesses of the samples affect this asymmetry phenomenon.

The authors do not show any characterization of the oxidized Fe₃GeTe₂ layers in the samples. Is the only the top layer oxidized or both the top and bottom layers are oxidized? What is the exact thickness and roughness of the oxidized Fe₃GeTe₂ layer? How about the pure Fe₃GeTe₂ thicknesses in your samples? Are there any changes of magnetic properties (such as magnetic moment of Fe, uniaxial anisotropy, and/or exchange stiffness) in near interface region where the oxidation of the near-surface layer of the Fe₃GeTe₂ takes place? How the authors may justify the statement based on the magnetic phase diagrams with different thicknesses when the states of the oxidized Fe₃GeTe₂ layers have not been checked? In Fig. 3 and Fig. 6, there are no information about the thicknesses of samples in the LTEM experiments and the micromagnetic simulations.

Overall, the work is valid and well organized, but I fail to see the conceptual novelty in the results. For the reasons above, I cannot support publication of this study in Nature Communications.

Reviewer #2:

Remarks to the Author:

The manuscript titled "History-dependent domain and skyrmion formation in 2D van der Waals

magnet Fe₃GeTe₂" by the authors of M. T. Birch et al. studied the phase diagram of Fe₃GeTe₂ films as a function of temperature and film thickness with zero-field cooling (ZFC), field cooling (FC) and field sweep (FC). The authors claim that the phase diagram is strongly dependent on the history of the applied field. This phenomenon actually is not very surprising because the applied field will change the magnetic order and reserve energy, it will surely be shown in the following measurement. However, the work is very well organized and written. Moreover, in the sense of the intensive study of skyrmions in 2D magnets, especially in the close to room temperature metallic 2D materials, FGT, the manuscript can be a good example of a systemic study of Skyrmions in FGT, it will be interesting for researchers in the skyrmiontronics field as well as 2D materials field. Therefore I am opting to publish the work after the authors address the following questions:

(1) In Figure 2, there are very few points in the SD regain, as a theorist, I am not familiar with the detailed measurement, however, it seems the points shown in the figure are too few. Is it difficult to have more points or those points are enough?

(2) Regarding the simulation part, the authors claim that the DMI is 0.12 mJ/m² for FGT samples. However, the samples they measured are 35, 50, and 60 nm, for different thicknesses, the micromagnetic DMI should have different amplitude.

(3) In addition, the DMI they got is just based on the micromagnetic simulation where the Bloch skyrmions transformed to Neel-type ones. However, this conclusion is not convincing at all. The authors should either measure the DMIs from experiments or calculate them from the first principles or else this part could not be accepted by readers in the future.

(4) Back to the history-dependent story, it will also work for other 2D magnets with inversion symmetry breaking, the authors may consider.

(5) In lines 101 and 104, there are some typos, two "where" and "fr" should be corrected.

Referee Response

We are pleased that both referees recognised the solid science and extensive analysis which has gone into our study. While one referee was positive about the manuscript, the other raised issues about the novelty of our findings in comparison to the existing literature. As well as highlighting where we believe the novelty in our results can be found, we have endeavoured to improve the manuscript in several aspects with respect to their comments. We thank the referees for their time reading the manuscript and helpful questions and suggestions.

Referee 1

1) *“The manuscript by M. T. Birch et al. reports the observation of spin textures for different magnetic fields and temperature histories in the layered ferromagnet Fe₃GeTe₂. The magnetic phase diagrams are systematically studied at varied magnetic fields and temperatures using scanning transmission x-ray microscopy (STXM) and Lorentz mode transmission electron microscopy (LTEM). The authors have demonstrated that spin textures such as uniformly magnetized, stripe domain, and skyrmion states can be selected at the same temperature and field point by the various sample thicknesses on the different measurement protocols. The major concern of the work is the stabilized spin textures by the sample history prior to measurements. The work is scientifically sound and the high-resolution magnetic imaging experiments are solid.”*

We are pleased that the referee recognises the detailed measurements and extensive analysis we have put into this study, and thank them for their time spent considering our work.

2) *“However, the work is somewhat consistent with previous studies [Refs. 22 and 37] that has found stripe domain or skyrmion structures in layered Fe₃GeTe₂ when the film reaches certain thickness.”*

It is correct that our work is not the first to present observations of skyrmions and stripe domains within FGT flakes. However, we believe our presented results remain novel in multiple aspects:

- A) The detailed and complete magnetic phase diagrams of FGT flakes for different sample histories and thicknesses have not been presented within the literature. Moreover, as we shall discuss further in answer to comment 4), the phase diagrams we present show significant differences from those of other known skyrmion hosts, including the B20 bulk chiral magnets, and chiral thin film systems.
- B) In particular, the result that all three magnetic states can be stabilised at the same applied field and temperature, which is featured in Figure 1, is a result so far unique to FGT flakes, and which we believe has not been explicitly demonstrated before. As far as we are aware, this behaviour has not been observed to occur within either bulk chiral magnets or chiral thin film systems. This result means that the spin texture within FGT may be more controllable in comparison to traditional skyrmion systems.
- C) While previous works have speculated that the out-of-plane anisotropy may be responsible for the temperature dependent size of stripe domain states within FGT, we believe our work is the first to directly demonstrate the resulting history-dependent effects. On this point, the contribution from our mean field simulations is also significant: the simulations are able to qualitatively reproduce the experimental magnetic phase diagram, and which we believe they are the first to simulate the full qualitative temperature-dependent magnetic behaviour of the experimental FGT sample. This was possible due to the ability to include temperature within the mean field model we

have utilised, which is known to be difficult to achieve with typical micromagnetic methods.

In addition, while we are pleased that both referees agree with our conclusions that the dipolar interaction plays a significant role in the formation of the spin textures in FGT, we note this matter is still quite controversial within the literature.

Two recent examples, both published since the initial submission of our work, put forward different viewpoints on this matter. The first work, based on simulations, argues that the observed Néel skyrmions are not stabilised by DMI at all, arguing instead that “fourth order interactions” are responsible [C. Xu, et al. *Advanced Materials*, 2107779 (2022)]. A second work [A. Chakraborty et al. *Advanced Materials*, 2108637 (2022)], attributes the observed thickness dependence of the observed spin textures to a DMI contribution, which we believe is rather a typical signature of dipolar-induced stripe/skyrmion states.

Both of these papers appear to make much mention of the dipolar interaction, highlighting the current conflict within the literature (indeed, from what we could see the words “dipolar”, “stray field” or “demag” do not seem to appear within the main text of either paper). From the perspective of future developments on FGT, we feel that it is vitally important to clear up this matter. If the further works on skyrmions in FGT proceed to explore how the properties of the skyrmions within FGT can be controlled by, for example, heterostructure stacking, or voltage/current effects, it is crucial that there is a fundamental understanding of the energy landscape which gives rise to the observed spin textures within FGT flakes, and their unique history dependent behaviours. Indeed, the controversy of this particular topic of dipolar vs DMI stabilised spin textures goes beyond the current FGT material, and similar disagreements exists around other skyrmion systems (for example, the antiskyrmion host $\text{Mn}_{1.4}\text{PtSn}$).

The existence of this controversy is the principle reason that we have gone to considerable effort to present multiple datasets demonstrating the dominant contribution of the dipolar interaction, while recognising that there must be some form of DMI present to realise the Neel-type structure. So, we would posit that while the referees may find it “expected” that the sample would behave this way, we believe that the results may not be so clear to others within the skyrmion field. Now that we have seen that the referees appear to agree with us on this matter, we have added some additional discussion to the manuscript to highlight these issues further.

3) *“In addition, the field cooling process (quenching process) could enhance skyrmion phase, as has already discovered in the conventional chiral-lattice magnets [Nat. Phys. 12, 62 (2015), Nat. Mater. 15, 1237 (2016), Nature 564, 95 (2018)] as well as the same layered ferromagnet (Fe₃GeTe₂) [Refs. 32-37].”*

The referee is correct that the field-cooling procedure to quench skyrmions to lower temperatures has been observed previously in bulk chiral magnets, in thin-film systems, and within the FGT system. However, we argue that the field stability of the skyrmions within FGT following this process has not been well-explored, and a similar field-cooled phase diagram, revealing the full extent of the skyrmion phase, has not previously been reported for FGT.

In this instance, the high negative fields (after field cooling in positive applied field) which our results show the skyrmions survive to, is indicative of their primarily dipolar-stabilised origin. As an example, we compare our result to a phase diagram of a typical bulk chiral magnet, showing how the skyrmions (labelled as SkL for skyrmion lattice) typically barely survive past 0 mT in a DMI-dominated system, in this case from a previous paper of ours on Cu_2OSeO_3 [Birch, et al. *Communications Physics* 4, (2021)]:

Our point here is that the exact nature of the phase diagram, and the extent of the skyrmion state, as well as being useful for exploring potential future works to control the spin texture, can reveal important clues about the contributions of magnetic interactions present in the sample, and therefore act as an essential foundation for future work manipulating the skyrmions within FGT.

4) *“So, what is the significant novelty of the history-dependent spin textures in layered Fe₃GeTe₂ compared to that of the conventional chiral-ferromagnetic multi-layers?”*

We feel this question reveals the most crucial points to establish the novelty of our work, so here we will compare our phase diagrams of FGT to three other related systems:

- 1) Chiral multilayer systems (interfacial DMI)
- 2) B20 chiral magnets (bulk DMI)
- 3) Dipolar stabilised bubble systems (no DMI).

Multilayer thin films:

We found that there are very few examples of magnetic phase diagrams which have been measured for chiral ferromagnetic multi-layer systems, likely because research on chiral multilayers hosting skyrmions has primarily focused on room temperature measurements with a focus on device applications. One example, from a paper by the co-author K. Litzius [I. Lamesh, et al. *Advanced Materials* (2018)], is a pictorial phase diagram of a typical multilayer thin film system, which we reproduce below. It is important to note here that this is a phase diagram where the authors performed a current pulse to nucleate the observed state for several different pulse heights U , and is therefore not directly comparable to our phase diagrams obtained with static, field dependent measurements.

There are two key points here, firstly field-temperature phase diagrams of chiral ferromagnetic multilayers are mostly unexplored within the literature. Secondly, while it has been established that both skyrmions and stripe states can be nucleated at 0 mT, it does not appear to be possible to stabilise the uniformly magnetised state at zero field (at least for the range of

temperatures explored in the previously mentioned publication), making the behaviour of FGT quite different.

B20 Chiral magnets:

The skyrmion pocket seen in the ZFC and FS phase diagrams of our FGT flake samples, which is confined to a small region close to T_C , is in itself quite interesting. As mentioned in the main text, it resembles the small skyrmion pocket typical of a bulk B20 chiral magnet.

The phase diagram on the right is a typical phase diagram of bulk B20 chiral magnet MnSi [Mühlbauer et al. *Science* **323**, 915-919 (2009), which also exhibits a small skyrmion pocket (known as the A-phase) close to T_C of the material.

This becomes more interesting when considering the magnetic phase diagram of thinned samples of B20 chiral magnets. On the order of the thickness of our FGT flakes, the skyrmion pocket typically expands with decreasing thickness, as shown by this phase diagram of a thin FeGe lamella [X. Yu, et al. *Nat. Mater.* **10**, 106-109 (2011)]. This is the opposite behaviour shown by the present measurements of our FGT flakes,

Thus, we contend that the magnetic phase diagram of our FGT flake sample is significantly different to that of bulk chiral magnets, which exhibit predominantly DMI-stabilised skyrmions. In these systems, once again it is sometimes possible to stabilise skyrmions in a metastable state at 0 mT, but it is not possible to realise the uniformly magnetised state at zero field – typically the helical state always stabilises at low fields.

Dipolar stabilised bubble systems:

A typical phase diagram from a dipolar-stabilised bubble skyrmion system, comparable to our ZFC or FS phase diagrams, is shown here [A. Kotani, et al. *Phys. Rev. B* **95**, 144403 (2017)]:

In these systems, non-chiral magnetic bubbles/skyrmions are exhibited in thin films/lamellae due to the dipolar interaction, and exhibit nonchiral Bloch-type winding due to the lack of interfacial DMI. The key difference in comparison to our samples is the large temperature extent of the bubble phase, where skyrmion bubbles can be formed from the stripes across a large range of temperatures, in comparison to the small skyrmion pocket observed in the FGT samples. Of course a key difference of FGT compared to these materials is that FGT presents monochiral spin textures, whereas skyrmions of both chiralities may exist within typical dipolar stabilised systems.

We will note here that the reference brought up by the referee, the work by H. Wang et al, does present magnetic phase diagrams of FGT flakes. However, we contend that they are incomplete and partially incorrect. Firstly, the phase diagram presented following the ZFC procedure indicates that Néel skyrmions are stabilised. We suspect that in reality there could have perhaps been a residual field in their instrument, which resulted in formation of skyrmions by field cooling, and therefore an incorrect ZFC phase diagram. Secondly, the results presented in that paper miss the existence of the small skyrmion pocket in both phase diagrams, which is revealed in our measurements. Thirdly, the paper does not present thickness dependent measurements of the phase diagram.

In summary, the phase diagrams we have presented in our work show new behaviour, which is significantly different from that seen in chiral multilayer, bulk B20 chiral magnets and dipolar stabilised non-chiral skyrmion systems. Moreover, the possibility to stabilise the Sk, SD and UM states at 0 mT across a wide range of temperatures appears to so far be unique to the Fe_3GeTe_2 system. Beyond our experimental determination of the magnetic phase diagrams, we contend that we have also directly demonstrated why the phase diagram of FGT appears this way: our successful qualitative reproduction of the magnetic phase diagram using the meanfield simulation method verifies that it is due to the strong temperature dependent anisotropy that this behaviour is exhibited. The simulations are able to reproduce the experimental magnetic field and temperature FS phase diagram.

We have added an additional paragraph to the section discussing the magnetic phase diagram results to summarise the comparison we have presented here, in order to better contextualise the different behaviour exhibited by the FGT magnetic phase diagrams (page 11 of the updated manuscript).

5) *“The authors also mentioned that the history-dependent spin textures based on the various sample thicknesses suggest that the interplay of the dipolar interaction is the primary stabilization of mechanism for the spin textures in the samples, despite the presence of the interfacial DMI. I think that the spin textures could be stabilized by the correlation among the dipolar interaction, the interfacial DMI, and the uniaxial anisotropy [Refs 49 and 50]. This correlation should be strongly affected by the sample thickness and history measurements.”*

We agree with the referee, and this is exactly the interpretation we have tried to present within the main text: the stripe and skyrmion domains are predominantly stabilised by the dipolar interaction and uniaxial anisotropy, but there must be some minor DMI contribution in order to twist the domain walls into the Néel-type configuration. Once again, we are glad that the referee agrees with us on this interpretation, and hope they would concur that a robust demonstration of this conclusion to clarify the current disagreement within the literature is useful. We have updated the text in several places to make this clearer (mainly in the discussion on page 20-21 of the updated manuscript).

6) *“However, there is a weak discussion on the mechanism of spin texture formations during the different field- and temperature-histories. The authors need to discuss in detail the comparison of the spin texture formation mechanisms, which have different thicknesses and measurement-histories. Moreover, it would be useful to discuss how dipolar interaction dominantly affect the spin texture during the field cooling process.”*

The referee is correct that we could have been clearer on these points. We have endeavoured to expand our explanation of these points, including two new points discussing the nucleation and formation of the stripe-like states via field-sweeping, and the dipolar-dominated formation of skyrmions by the field-cooling process (pages 11-13, and page 15 of the updated manuscript).

7) *“I also wonder what role the interfacial DMI has on the observed spin textures for different field and temperature histories.”*

We suspect that the dominance of the dipolar stabilisation mechanism means that the comparatively small DMI plays a rather minor role in the overall behaviour of the sample, beyond fixing the helicity of the domain walls to be Néel-type. We have updated the discussion section to touch upon this point (page 20-21 of the updated manuscript).

8) *“The thickness- and history-dependences of micromagnetic simulations can be considered as a convincing argument supporting the role of dipolar interaction in the layered Fe₃GeTe₂.”*

We agree with the referee here, this is exactly what we were trying to convey in our arguments. We have updated the discussion section to be more direct and clear on this point. (page 18 of the updated manuscript).

9) *“Furthermore, what is the origin of the asymmetry in the formation of the skyrmion states via the field cooling process as shown in Fig. 2? I also wonder how the thicknesses of the samples affect this asymmetry phenomenon.”*

We agree with the referee that as originally presented this might be confusing. In fact, the origin of the asymmetry is only because we are showing a magnetic phase diagram for cooling with an applied positive field. If we were cooling with a negative field, the phase diagram would be mirrored around the x-axis. It's the same for the field-sweep phase diagram – it would be mirrored if we had swept from positive to negative field rather than negative to positive. We have made a note of this in the manuscript to make this point clearer (page 10 of the updated manuscript).

10) *“The authors do not show any characterization of the oxidized Fe₃GeTe₂ layers in the samples. Is the only the top layer oxidized or both the top and bottom layers are oxidized? What is the exact thickness and roughness of the oxidized Fe₃GeTe₂ layer? How about the pure Fe₃GeTe₂ thicknesses in your samples?”*

We have estimated the thickness of the FGT and its oxide layer using two methods, as presented in the main text and in Supplementary Note 3. Firstly, we estimated that since the FGT thicknesses with 15 nm and below thicknesses showed no ferromagnetism, we can assume that these are almost completely oxidised, thus suggesting that somewhere around 7-10 nm of both surfaces of the sample are oxidised. This is corroborated by our XMCD spectra analysis, presented in the supplementary information, where the Fe magnetic moment measured in the (oxidised) flake sample at the 60 nm thickness was about 67% of that measured from the spectra measured on the unoxidised, cleaved bulk sample. This indicates that 33% of the Fe in the FGT flake was no longer magnetic, so we can assume about 33% of the 60 nm thickness region was oxidised, or around 10 nm of both the top and bottom surface.

Nevertheless, the referee is correct that these are only approximations. Therefore, we decided to perform TEM measurements of a comparable FGT flake sample. The original sample was prepared on a Si₃N₄ membrane, which would have made extracting a lamella suitable for TEM observations quite challenging. Therefore, we created a similar sample but instead exfoliated on a SiO₂ substrate, following the same preparation procedure where the sample spent a similar amount of time in ambient conditions. The results reveal the thickness of the oxide layer following our method is approximately 7 nm, agreeing well with our previous estimation of the FGT flake investigated by STXM. As well as mentioning this in the main text, we have added an additional paragraph to Supplementary Note 3, and additional supplementary Fig. S7, to present the results of this TEM experiment.

11) *“Are there any changes of magnetic properties (such as magnetic moment of Fe, uniaxial anisotropy, and/or exchange stiffness) in near interface region where the oxidation of the near-surface layer of the Fe₃GeTe₂ takes place? How the authors may justify the statement based on the magnetic phase diagrams with different thicknesses when the states of the oxidized Fe₃GeTe₂ layers have not been checked?”*

Since our x-ray microscopy technique is a transmission method, we were unfortunately unable to gain information about any depth dependence of the magnetic properties. However, the spectra presented in the supplementary information reveal significant information about the oxide formation, which can be determined from the valence change of the Fe atoms shown in the spectra. We have used this information to judge the oxide content within our flake samples. The lack of XMCD signal on the secondary, oxidised Fe L₃ peak is a very strong indication that this valence state does not exhibit any ferromagnetism, and thus the magnetic state of the oxidised layer should not be a consideration.

Furthermore, we would argue that since the oxidisation occurs from the surface of the flake, we expect the total oxide thickness, and its roughness, should be largely similar across all thickness regions of the FGT flake. All presented phase diagrams are from the same FGT flake, which could in turn be expected to possess an oxidised FGT layer on each surface of the same thickness. As shown in Fig 3, we recorded the microscopy data used to plot the magnetic phase diagrams by imaging all three thickness regions simultaneously, so all phase diagrams presented are from regions with a similar oxidised FGT layer.

We further suggest that the thickness of the oxide layer should have little effect on the magnetic properties, since effects such as any interfacially-induced DMI would typically be determined by the interface of the two layers. Thus, we speculate that additional oxide thickness beyond 3-4 nm may not significantly alter any interfacial interaction (although again

we note that we do not have direct evidence as to the origin of the DMI, and this seems to remain an open question in the literature, although the thickness dependence of the magnetic textures – ie the Bloch to Neel thickness transition – does indicate it to be related to the surface of the FGT flake).

12) *“In Fig. 3 and Fig. 6, there are no information about the thicknesses of samples in the LTEM experiments and the micromagnetic simulations.”*

The referee is correct that we have overlooked these points. We have included the thickness of the LTEM sample in the methods section and the supplementary information, and the full dimensions of the micromagnetic simulations (pages 24-26 of the updated manuscript)

13) *“Overall, the work is valid and well organized, but I fail to see the conceptual novelty in the results. For the reasons above, I cannot support publication of this study in Nature Communications.”*

Once again, we would like to thank the referee for their appraisal of our study, and for their detailed comments and feedback. Following our changes to the manuscript, we believe that the work has been greatly improved. Moreover, by comparison to previous studies on other skyrmion systems, we have demonstrated the novelty of our results, which we believe will be well received by the skyrmion community, and those interested in 2D magnetic for spintronics, and hope that the referee can now find our work worthy of publication.

Referee 2

“The manuscript titled “History-dependent domain and skyrmion formation in 2D van der Waals magnet Fe₃GeTe₂” by the authors of M. T. Birch et al. studied the phase diagram of Fe₃GeTe₂ films as a function of temperature and film thickness with zero-field cooling (ZFC), field cooling (FC) and field sweep (FC). The authors claim that the phase diagram is strongly dependent on the history of the applied field. This phenomenon actually is not very surprising because the applied field will change the magnetic order and reserve energy, it will surely be shown in the following measurement. However, the work is very well organized and written. Moreover, in the sense of the intensive study of skyrmions in 2D magnets, especially in the close to room temperature metallic 2D materials, FGT, the manuscript can be a good example of a systemic study of Skyrmions in FGT, it will be interesting for researchers in the skyrmionics field as well as 2D materials field. Therefore I am opting to publish the work after the authors address the following questions:”

We thank the referee for their kind consideration of their work, and their suggestion to publish the work, following our answers to their questions.

1) *“In Figure 2, there are very few points in the SD regain, as a theorist, I am not familiar with the detailed measurement, however, it seems the points shown in the figure are too few. Is it difficult to have more points or those points are enough?”*

The referee is correct that we would have liked to include more points in the low temperature region, to better characterise the extent of the stripe domain state. However, due to the limited amount of time available in synchrotron-based experiments (typically at BESSY II, each experiment is limited to 6x12 hour shifts within a week, and each group may not get any more time for another 6 months), we decided to prioritise obtaining more temperature points in the higher temperature regions of the phase diagram, around the skyrmion pocket. As for the validity of mapping the stripe domain state with only points between every 25 K between 125 and 175 K, we believe that there is unlikely to be a significant change of behaviour between these temperatures.

2) *“Regarding the simulation part, the authors claim that the DMI is 0.12 mJ/m² for FGT samples. However, the samples they measured are 35, 50, and 60 nm, for different thicknesses, the micromagnetic DMI should have different amplitude.”*

The referee is correct that the strength of the DMI, if it indeed comes from the interface of the FGT with an oxide layer, may possibly change for the different thicknesses of the experimental FGT flake. The rest of our answer to this comment is included in the response for the following comment.

3) *“In addition, the DMI they got is just based on the micromagnetic simulation where the Bloch skyrmions transformed to Neel-type ones. However, this conclusion is not convincing at all. The authors should either measure the DMIs from experiments or calculate them from the first principles or else this part could not be accepted by readers in the future.”*

We agree with the referee that the value of 0.12 mJ/m² should not be taken as the true value of the DMI present in the experimental sample. Rather, as we noted within the main text, we wanted to present a rough lower bound of the DMI within the sample, and highlight that a large DMI value is not required in order to realise Néel-type skyrmions within a predominantly dipolar-stabilised system. As presented originally, this was perhaps not as clear as it should have been. Therefore, we have updated the main text to include some more discussion of this point, and highlight that the particular value will depend on additional factors such as the thickness of the sample, and the exact micromagnetic parameters utilised in the simulations, and to emphasise that the presented value is only an estimation (page 18). True determination

of the DMI value would require, for example, Brillouin light scattering measurements, to which we do not have any access or expertise, and would suggest are beyond the scope of the present work.

4) *“Back to the history-dependent story, it will also work for other 2D magnets with inversion symmetry breaking, the authors may consider.”*

This is certainly a useful point that we could have included in the discussion of our results. We believe that it is the strong temperature dependent anisotropy which is primarily responsible for the strongly history dependent phase diagram of FGT. It's possible that similar properties may well be found in other 2D magnets, or otherwise realised by heterostructure stacking or for example gate voltage control of the magnetism. We have updated the discussion section to touch upon these points (page 20-21).

(5) *“In lines 101 and 104, there are some typos, two "where" and "fr" should be corrected.”*

We thank the referee for catching these mistakes, which have now been corrected, and for their time spent preparing their valuable feedback on our work. We hope that our responses to these comments and updates to the manuscript are well received.

Reviewers' Comments:

Reviewer #1:

Remarks to the Author:

In the revised manuscript, the authors addressed most of concerns raised by the referees. Although I still have ambiguities about the thickness dependent mechanism of spin texture formation during field- and temperature-histories, the authors have shown its potential. Considering their reply to the comments and questions of the referees, I can now recommend the publication in Nature Communications.

Reviewer #2:

Remarks to the Author:

I thank the authors for addressing my concerns very seriously. I am satisfied with all their answers. One more suggestion is that the authors should include more references regarding the DMI study using 2D magnets, such as Nano Lett. 22, 2334 (2022) and Phys. Rev. B(R) 102, 220409 (2020) etc.. So that the scope of the study is not limited to one 2D magnet.

Referee 1

“In the revised manuscript, the authors addressed most of concerns raised by the referees. Although I still have ambiguities about the thickness dependent mechanism of spin texture formation during field- and temperature-histories, the authors have shown its potential. Considering their reply to the comments and questions of the referees, I can now recommend the publication in Nature Communications.”

We thank the referee for their appraisal of the new version of the manuscript. Along the lines of the referee’s final comment, we have included an additional sentence to the discussion section, pointing out that investigation a wider range of thicknesses may reveal more about the role of thickness in the spin texture formation following different sample histories, and whether skyrmions may be found at the single layer limit.

Referee 2

“I thank the authors for addressing my concerns very seriously. I am satisfied with all their answers. One more suggestion is that the authors should include more references regarding the DMI study using 2D magnets, such as Nano Lett. 22, 2334 (2022) and Phys. Rev. B(R) 102, 220409 (2020) etc.. So that the scope of the study is not limited to one 2D magnet.”

The referee’s suggestion to widen the scope of the DMI discussion to include new 2D magnets is a nice one. The suggested new 2D material structures in the Nano Lett. paper, showing symmetry groups with an inherent chirality, and therefore DMI, we found particularly interesting, and had overlooked before. We have added additional sentences to the discussion section to discuss these ideas.